# The Registration Situation and Use of Mycopesticides in the World

**DOI:** 10.3390/jof9090940

**Published:** 2023-09-16

**Authors:** Yali Jiang, Jingjing Wang

**Affiliations:** 1College of Plant Protection, South China Agricultural University, Guangzhou 510642, China; jiangyali@stu.scau.edu.cn; 2College of Horticulture, South China Agricultural University, Guangzhou 510642, China

**Keywords:** mycopesticides, history, application, trend of development

## Abstract

Mycopesticides are living preparations that use fungal cells, such as spores and hyphae, as active ingredients. They mainly include mycoinsecticides, mycofungicides, mycoherbicides and nematophagous fungi. The utilization of fungi for controlling agricultural pests can be traced back to approximately 1880, when entomopathogenic fungi were initially employed for this purpose. However, it was not until 1965 that the world’s first mycopesticide, *Beauveria bassiana*, was registered as Boverin^®^ in the former Soviet Union. In past decades, numerous novel mycopesticides have been developed for their lower R&D costs, as well as the environmentally friendly and safe nature. In this review, we investigated the mycopesticides situation of registration in USA, EU, China, Canada and Australia. Superisingly, it was found that the registered mycopesticides are extremely raised in recent years. Currently, the insecticides, fungicides (nematocides) and herbicides were respectively registered 27, 53 and 8 fungal strains. This paper also analyzes the main problems currently faced by mycopesticides and offers suggestions for their future development.

## 1. Introduction

With the rapid growth of the global population, food production faces significant challenges. The era of organic synthetic pesticides began in the early 1940s in response to the increasing demand for food. Initially, organochlorine insecticides like DDT were introduced, followed by the widespread use of chemically synthesized pesticides, which greatly enhanced the human ability to control agricultural pests. This led to increased crop production, addressing the food needs in most regions, and ensuring farmers’ income [1]. However, the extensive use of chemical pesticides has had a significant impact on the environment [2], as well as non-target organisms [3], and has resulted in the escalating issue of pesticide residues [4]. Moreover, the excessive use of chemical pesticides has led to over 500 species of pests developing resistance to one or more insecticides [5]. To mitigate reliance on chemical pesticides, various countries have implemented policies to encourage the development of biological pesticides. For instance, the Environmental Protection Agency (EPA) of the United States established the Biopesticide and Pollution Prevention Division (BPPD) in 1994 to streamline the registration of biopesticides. In 2009, the European Union (EU) introduced the ‘Sustainable Use of Pesticides Directive’ advocating for biological control. Brazil established the Brazilian Association of Biocontrol Companies (ABCBio) in 2007 to promote biological control [6]. In 2015, the Ministry of Agriculture and Rural Affairs of China formulated and issued the Action Plan for Zero Growth in Pesticide Use by 2020. As of September 2022, China has completely banned the use of 48 chemical pesticides, with 21 chemical pesticides being prohibited in specific regions [7].

Compared with chemical pesticides, biopesticides are environmentally friendly, safe for non-target organisms, and less likely to develop resistance [8]. The biopesticide industry has been experiencing significant growth since 2010, with a compound annual growth rate (CAGR) of 10–20% per year [9]. As of August 2022, there are 576 biopesticide products registered, of which 65% are microbial pesticides [9]. The development and market application of a novel chemical pesticide require an investment of approximately $250 million and a minimum of 10 years. On the other hand, a new microbial pesticide only requires an investment of $1–2 million and can enter the market within 3–5 years [10,11]. Reports had indicated that there are currently around 175 active substances of microbial pesticides available for agricultural production [12]. Microbial pesticides dominate the markets in Latin America and North America, while their presence in the EU market is relatively small due to strict regulatory policies [9]. Microbial pesticides encompass bacterial biopesticides, fungal pesticides, viral pesticides, actinomycetes, protozoa, and other types. In 2016, mycopesticides accounted for 10% of the global biopesticide market [13]. Mycopesticides primarily utilize fungal conidia as the main active ingredient. Mycoinsecticides that occupy major markets include *Beauveria bassiana*, *Metarhizium* spp., and *Akanthomyces lecanii*. Mycofungicides, on the other hand, consist of *Trichoderma* spp., *Ampelomyces quisqualis*, *Paraphaeosphaeria minitans*, *Gliocladium* spp., and there are also nematophagous fungi such as *Purpureocillium lilacinum* [14,15]. Mycoherbicides, however, make up only a small fraction of the biopesticide market.

Unlike bacteria and viruses, which primarily invade through the digestive tract, entomopathogenic fungi directly penetrate the cuticle of insect, exerting a strong contact effect. The main mechanism of action for entomopathogenic fungi in controlling arthropod pests involves conidia attaching to the insect cuticle; germinating to form germ tubes or appressorium; releasing proteases, chitinases, and lipases to penetrate the host cuticle; entering into the insect hemocoel; and reproducing in large numbers after successful colonization. Additionally, they produce toxins to kill the host. Finally, the fungus breaks out of the epidermis and produces conidia again to infect other insects [16,17,18]. Mycoinsecticides are more effective against certain piercing-sucking pests. The mechanism of action of mycopesticides involves direct or indirect effects, primarily through the production of metabolites or antibiotics that inhibit pathogens, mycoparasitism, competition for nutrients and sites, induction of plant resistance to pathogens, and promotion of plant growth [19]. Most mycoherbicides show strong host specificity. However, mycopesticides have some disadvantages, including an unstable control effect, slow action, and susceptibility to environmental factors, which limit their development.

## 2. Registration of Mycopesticide Products 

In the United States (US), the EPA has less stringent data requirements for registering biopesticides compared to traditional pesticides. Additionally, the review time for biopesticides by the EPA is shorter. Mycopesticide products were registered as early as 1981, with three products of *Nosema locustae* registered in 1980. The number of mycopesticide products is the highest in the US compared to other countries. The registration of plant protection products in the EU is carried out according to the rules of Regulation 1107/2009. The average time for microbial biological control agents (MBCA) authorization and microbial biological control products (MBCP) approval improved from 1845 days under Directive 91/414/EEC to just 1369 days under the new Regulation (EC) 1107/2009.The approval process for MBCA is conducted at the EU level, while the approval of MBCP is done at the national level. The registration of MBCPs in the EU is more complex due to different processes at the EU and Member State (MS) levels, large actor heterogeneity, and low flexibility [20]. The number of MBCA approvals in the EU has steadily increased since 2013, but most approved strains result in MBCP were submitted for approval in only a few member states [21]. The registration of fungal pesticide products in China started relatively late, and the number of registered strains is small. Currently, only 16 fungal strains, mainly entomopathogenic fungi and mycofungicides, are registered in China. There are no fungal varieties registered for controlling weeds. However, as of 2022, the number of product registrations is increasing and approaching the level seen in the US. In 1992, Canada registered BioMal^®^, the first fungal herbicide product containing *Colletotrichum gloeosporioides* f.sp. *malvae* as its active ingredient. However, due to its limited market size, the product was discontinued after 2 years [22]. Since then, Canada has approved a total of 28 fungal strains and 88 fungal products for registration between 1992 and 2022. In 1996, *Metarhizium*-based products BioGreen^®^ were registered for the first time as fungal biopesticides in Australia to combat pests including canegrubs, termites, and locusts. Registration of pesticides is governed by the Agricultural and Veterinary Chemicals Code Act 1994 and administered by the Australian Pesticides and Veterinary Medicines Authority (APVMA). Small projected returns and lengthy registration procedures are expected to limit the registration of microbial pesticides in Australia. The registration of a microbial pesticide requires the assessment of a comprehensive set of data on toxicology, efficacy, storage and field residues. As Australia is an independent island nation, the assessment of harmful effects of a microbial pesticide on local species before the introduction of a new microorganism [23,24]. The development of the registration of mycopesticides in the US, China, Canada, Australia, and the EU over the last 30 years is shown in Figure 1.

## 3. The Development History and Application of Mycopesticides

### 3.1. Mycoinsecticides

According to incomplete statistics, there are over 750 species and 100 genera of fungi that can infect insects [25]. The medicinal value of white muscardine silkworm was recorded 2000 years ago in the agricultural monograph Shennong Bencao Jing, which is the earliest human record of the fungal infection of insects [26].The earliest record of entomopathogenic fungi in Europe can be traced back to 1779 when DeGeer described flies infested by *Entomophthora muscae* [27]. In 1835, the Italian scientist Agostino Bassi discovered that a large number of silkworms, *Bombyx mori*, were covered with white powder, and identified the pathogenic agent of white muscardine disease in silkworms. He also proposed that this fungus could be used to infect silkworms and other species, marking the first report of microorganisms being used to control pests [28]. In 1879, Elie Metchnikoff identified the pathogenic fungus of the wheat cockchafer *Anisoplia austriaca as Entomopthora anisopliae*, now known as *Metarhizium anisopliae* [29]. In 1888, Krassilstschik achieved the first industrial production of *M. anisopliae* in Russia for controlling the sugar beet weevil *Bothynoderes punctiventris* Germar, marking the first large-scale application of biological control in the world [30]. In 1965, the former Soviet Union approved the registration of Boverin^®^, a fungal insecticide based on *B. bassiana*, for controlling the Colorado potato beetle *Leptinotarsa decemlineata* and the codling moth *Cydia pomonella* [31]. In 1981, the US registered the first fungal pesticide under the trade name Mycar^®^, *Hirsutella thompsonii* Fisher, for controlling the citrus rust mite *Phyllocoptruta oleivora* Ashmead [32]. The registration and target of mycoinsecticides in the world are shown in Table 1. 

#### 3.1.1. *Beauveria*

There has been a total of 171 different types of fungal insecticides registered worldwide, with *Beauveria* being the most common, accounting for 58 types and 33.9% of the total [33]. *B. bassiana* is a widely distributed entomopathogenic fungus found in soil and is extensively utilized as a fungal insecticide. It has a broad range of hosts and can parasitize over 700 different insect species [34]. *Beauveria* insecticide products, consisting of 14 strains, have been registered in China, Canada, Australia, the EU, and the US. These products are primarily used for controlling Hemiptera, Lepidoptera, and Coleoptera pests. *B. bassiana* is used for large area control of pine caterpillars in China [35]. In Brazil, *B. bassiana* has been successfully employed to control whiteflies and coffee cherry beetles in large areas [6]. As early as 1999, the US registered the *B. bassiana* product ‘Mycotrol’ for the control of forestry and agricultural pests, including grasshoppers, sandflies, thrips, aphids, and others [36]. Another related species, *B. brongniartii*, has been used in Europe to control the European cockchafer *Melolontha melolontha* [37]. *B. brongniartii* products have been registered in Switzerland, Italy, and Austria [38].

#### 3.1.2. *Metarhizium*

*Metarhizium*, a member of the Ascomycota phylum, is known for its parasitic ability on over 200 insects, nematodes, and mites from 8 different orders. It is commonly used for controlling various agricultural and forestry pests including locusts, cockroaches, termites, rice planthoppers, and *Spodoptera litura* [39]. *M. anisopliae* has been utilized as a biological control agent for a long time, particularly in Brazil where it has been effective against spittlebugs in sugarcane [26]. Additionally, *M. acridum* has been extensively produced to combat locusts. Notably, Green Muscle^®^, developed by CABI Bioscience, has been successfully registered and implemented for production in Africa, where it is widely employed for controlling desert locusts *Schistocerca gregaria* [40].

#### 3.1.3. *Cordyceps*

Initially proposed by Persoon as *Isaria*, the entomogenous fungi of *Isaria* were classified in the genus *Paecilomyces* by Brown and Smith in 1957 [41], only to be reverted back to *Isaria* in 2005 [42]. In 2017, Kepler conducted a phylogenetic analysis of *Cordyceps*, analyzing 5 nuclear gene fragments, and classified most species in the *Isaria* family as *Cordyceps* [43]. Some commonly known species include *C. farinosa*, *C. fumosorosea*, *C. javanica*, *C. tenuipes*, and *C. cateniannulata*, among others. *C. fumosorosea* was registered in Japan in 2001 as a product preparation for the control of whiteflies and aphids [44]. *C. fumosorosea* currently has several products registered in the US, the EU, and Canada to control insect pests such as spider mites and whiteflies. Additionally, *C. farinosa* has a wide range of host species, especially lepidoptera, but there is no commercial product registration at present [45]. *C. javanica* has dual control effects on aphids and fungal diseases [46]. *C. tenuipes* exhibits significant pharmacological and medicinal effects. These effects include anti-tumor, anti-bacterial, anti-depressant, hypoglycemic, and hypolipidemic properties, as well as the ability to scavenge free radicals [47,48]. Additionally, *C. cateniannulata* has been found to effectively control various pests such as *Tetranychus urticae* Koch [49], aphid, nematode [50], and *Resseliella odai* [51].

#### 3.1.4. *Akanthomyces lecanii*

The genus *Akanthomyces* was proposed by Lebert in 1858. *Lecanicillium lecanii* was regarded as *A. lecanii* in 2017 [43]. *A. lecanii* was first discovered by Nivter in Ceylon (now Sri Lanka) in 1861 [52]. Due to its specific humidity requirements, *A. lecanii* commercial products are primarily used for controlling greenhouse pests. It is known for its ability to parasitize *Lecani coffeae* and has shown promising control efficacy against greenhouse pests such as aphids, thrips, whiteflies, and pest mites [53,54,55,56]. Additionally, it has the capability to parasitize certain plant pathogens like powdery mildew and rust fungus [57]. The safety evaluation of *A. lecanii* was completed in the 1970s by the United Kingdom, leading to its commercial production. The fungus has been formulated into products such as ‘Vertalec’ for aphid control and ‘Mycotal’ for whitefly and thrip control in greenhouses. These products have been registered in Denmark, Finland, The Netherlands, Norway, and the United Kingdom [36]. 

#### 3.1.5. *Hirsutella thompsonii*

In the 1950s, Fisher discovered a fungus known as *H. thompsonii* that could infect the citrus rust mite *Phyllocoptruta oleivora* Ashmead [58]. Subsequently, applied research on the fungus has been carried out by the US, Israel, and China. In 1972, the Citrus Research Institute of the Zhejiang Academy of Sciences in China successfully isolated *H. thompsonii* from citrus rust mites [59]. By the late 1970s, this fungus was processed into powder, which effectively controlled citrus rust mites. *H. thompsonii* is a significant parasite for various types of mites. India utilizes it to control coconut mites, while the United States employs it to control citrus rust mites *P. oleivora* and two-spotted spider mites *Tetranychus urticae* [60,61].

### 3.2. Mycofungicides and Nematophagous Fungi

In 1874, Roberts first demonstrated that *Penicillium glaucum* and bacteria had microbial antagonistic action in liquid media, introducing the term “antagonism”. In 1921, Hartley conducted an experiment to control the blight caused by Pythium by introducing 13 fungi with antagonistic potential into the soil. This marked the first attempt to use fungi to combat plant pathogens [62,63]. In 1932, Weindling demonstrated the biological control activity of *Trichoderma* against *Rhizoctonia solani*, thus recognizing the potential application of known fungal antagonists in plant disease control [64]. Subsequently, the inhibitory effects of the same species of *Trichoderma* against *Phytophthora*, *Pythium*, *Rhizopus*, and *Sclerotia* were observed. In 1928, Fleming’s discovery and purification of penicillin, along with its use in medicine, greatly accelerated the research on antagonists of plant pathogens [62]. The registration and target of mycofungicides or nematophagous fungi are shown in Table 2. 

#### 3.2.1. *Trichoderma*

In 1794, Peron first proposed *Trichoderma* spp. *Trichoderma* has been known since the 1930s for its ability to control plant pathogens and can be isolated from almost all soils containing vegetation. *Trichoderma* are typically anaerobic, facultative, and cosmopolitan fungi [65,66]. *Trichoderma* not only effectively controls plant pathogenic fungi but also enhances plant disease resistance, promotes plant growth and reproduction, modifies the rhizosphere environment, and facilitates nutrient absorption [67,68]. The mechanisms employed by *Trichoderma* to combat phytopathogenic fungi include competition, colonization, antibacterial activity, and direct fungal parasitism [69]. Common species of *Trichoderma* include *T. harzianum*, *T. viride*, *T. koningii*, *T. lignorum*, *T. hamatum*, *T. longibrachiatum*, *T. polysporum*, and *T. virens*. *Trichoderma* is known to parasitize at least 18 genera and 29 types of plant pathogens, including *Pythium* spp., *Sclerotinia* spp., *Verticillium* spp., *Fusarium* spp., *Botrytis cinerea*, and *Rhizoctonia solani* [70]. Currently, there are over 50 *Trichoderma*-based agricultural products available worldwide [71]. In countries like Brazil and other Latin American nations, *Trichoderma* is extensively utilized as a biocontrol agent for plant diseases. It is commonly used in seed treatment to manage seed and soil pathogens and to enhance the growth of various agricultural crops [72,73].

#### 3.2.2. *Ampelomyces quisqualis*

In 1852, Cesati first discovered that *Ampelomyces quisqualis* was a parasite of powdery mildew [74]. This fungus is capable of parasitizing over 65 species from 9 different genera within the powdery mildew. The *Ampelomyces* strain AQ10 or M-10, which was isolated from an *Oidium* sp. infecting *Catha edulis* in Israel, has been registered in the US and the EU as an active ingredient in the AQ10^®^ biofungicide product. The product is used specifically to control powdery mildew in various crops, particularly in grapes. Another biofungicide product, Q-fect^®^, has an active ingredient of *Ampelomyces* strain 94,013 that was isolated from *Podosphaera Phaseolus* on *Phaseolus angularis* in Korea. This product is primarily used for controlling cucumber powdery mildew in Korea [75,76].

#### 3.2.3. *Paraphaeosphaeria minitans*

*Coniothyrium minitans* was initially discovered by Campbell in 1947 on the parasitized sclerotia of *Sclerotinia sclerotia* in the US, and it has since been observed on all continents except South America [77,78]. Based on analyses of concatenated internal transcribed spacer regions of the nrDNA operon (ITS), large subunit rDNA (LSU), gamma-actin, and beta-tubulin gene sequences, *C. minitans* was reclassified as *Paraphaeosphaeri minitans* [79]. The application of *P. minitans* can be categorized into two approaches: soil application to minimize the amount of sclerotia inoculum, and spraying spores on diseased plants or crops to disinfect them. Numerous studies have reported the ability of *P. minitans* to infect and degrade sclerotia present in the soil [80]. There are registered products in the US, EU, Canada, and China for the prevention and treatment of *Sclerotinia*.

#### 3.2.4. *Paecilomyces*

The form genus *Paecilomyces* was first established by Bainier in 1907 with the description of a single species, *P. varioti*. Since then, several researchers including Thom (1910), Westling (1911), Sopp (1912), Zaleski (1927), Raper & Thom (1949), and Brown & Smith (1957) have conducted extensive research on this genus. In 1974, Samson described the morphology of the genus in detail and divided it into different species [41]. Some species in *Paecilomyces* retain their original genus, and some have been reclassified to other genera. For example, *P. lilacinus* (Thom) Samson has been assigned to the genus *Purpureocillium*, and *P. fumosoroseus* and *P. farinosus* were assigned to the genus *Cordyceps*. Within the genus *Paecilomyces*, *P. varioti* is not only effective against a variety of phytopathogenic fungi such as *Pythium spinosum* [81], *Fusarium oxysporum* [82], and *Phytophthora cinnamomic* [83], but also a potent nematophagous fungus, especially against the root-knot nematode *Meloidogyne* spp. [84,85]. Of course, *Purpureocillium lilacinum* is one of the most potential nematophagous fungus and can control various nematodes in different crops, although it is no longer in the genus *Paecilomyces* [86].

### 3.3. Mycoherbicides

The use of *Fusarium oxysporum* fungus in Hawaii during the 1940s to suppress the tree cactus *Opuntia megacantha* was the first attempt at using fungi to manage weed infestations. Although this endeavor was unsuccessful, it paved the way for future research [87,88]. Another notable example occurred in the 1960s when the US effectively controlled the persimmon trees *Diospyros virginiana* using hyphomycetous fungus *Acremonium diospyri* [89]. A highly successful case took place in Australia in 1971, where *Puccinia chondrillina* was employed to manage the rush skeleton weed *Chondrilla juncea* [90,91]. In 1981, the world’s first fungal herbicide, DeVine, was registered in the US. DeVine is a suspension of chlamydospores from the pathogenic strain of *Phytophthora palmivora*, which is utilized to control milkweed vine *Morrenia odorala* in citrus orchards, with a control efficacy of over 90% [92]. In the 1960s, China isolated the diseased soybean dodder *Cuscuta australis* and obtained *Colletotrichum gloeossporioides* f. sp. *cuscuata* ‘Lubao No. 1’, which proved to be highly effective in controlling soybean dodder and has since been widely applied in production [93]. The registration and target of mycoherbicides are shown in Table 3. 

#### 3.3.1. *Phytophthora palmivora*

In 1981, DeVine, a fungal herbicide, was registered in the US by Abbott Laboratories. This suspension was prepared from chlamydospores of *Phytophthora palmivora* and was the first fungal herbicide to be registered globally. It was primarily used for spraying on citrus orchards to control weeds [94].

#### 3.3.2. *Colletotrichum gloeosporioides*

*Colletotrichum gloeosporioides* f. sp. *aeschynomene* strain ATCC 20358 is a fungus known for its herbicidal activity on leguminous plants and rice fields. In 1982, it was approved by the US as an active ingredient in the product Collego^®^, which is used for controlling northern jointvetch *Aeschynomene virginica* [95]. Additionally, *Colletotrichum gloeosporioides* f.sp. *malvae* spores were registered in Canada in 1992, making them the first fungal herbicides to be registered in Canada for controlling round-leaved mallow *Malva pusilla* [96].

#### 3.3.3. *Chondrostereum purpureum*

*Chondrostereum purpureum*, a widely distributed fungus in deciduous trees in northern temperate regions, invades trees through fresh wounds [97,98]. It develops in the xylem of infected broad-leaved trees and shrubs, where the fungus plugs the xylem vessels, causing the cambium to die, rot, and discolor the wood center. This ultimately leads to plant wilting [99,100]. Additionally, the fungus produces a specific enzyme called endopolygalacturonase (endoPG), which moves to the leaves and causes silvery-gray symptoms, resulting in silver leaf disease in orchard trees [101,102]. The fungus has been developed as a biocontrol agent in North America, various European countries, and New Zealand to manage broadleaf weed trees in coniferous forests [103,104].

## 4. Problems and Development Trend of Mycopesticides

### 4.1. Problems

#### 4.1.1. Environmental Limitations

Ultraviolet light is a common stressor in outdoor environments. When fungi are exposed to ultraviolet irradiation, they experience DNA damage and produce cyclobutane pyrimidine dimers and pyrimidine–pyrimidone photoproduct These photoproducts can be cytotoxic and lead to gene mutations, growth defects, and cell death. While filamentous fungal cells have a DNA photolyase to repair damaged cells under visible light, their activity is diminished under strong light conditions [105,106,107]. Temperature is another crucial factor that affects the effectiveness of fungal pesticides in the field. The optimal temperature for the germination and growth of entomopathogenic fungi is between 23 and 28 °C [108]. Additionally, humidity plays a role in conidia germination. For example, it took 20 h to germinate *B. bassiana* at 25 °C and 95.5% relative humidity, whereas it took 72 h to germinate at 90% relative humidity [109].

#### 4.1.2. Virulence

Virulence is a crucial parameter for evaluating the effectiveness of fungal control. After successive subculture, strains often encounter the issue of degeneration, resulting in a decline in desirable traits and a significant decrease in virulence [110,111]. Continuous cultivation resulted in *Aspergillus flavus*-reduced spore production, the proliferation of aerial hyphae and inability to produce sclerotia [112,113,114]. The pathogenicity of strains can also be influenced by factors like medium composition, environmental conditions, and contamination [115]. *B. bassiana* showed decreased virulence to *Dociostaurus maroccanus* after two passages on the sabouraud dextrose agar medium but increased virulence after two subcultures on malt agar medium [116].

#### 4.1.3. Difficulties in Promotion

Fungal pesticides are living preparations that have a short storage time, slow effect in the field, and high prices. These factors present challenges for fungal pesticides in terms of the market. The effect of fungi on target insects takes about 3–15 days, and a large number of fungal spores is required to effectively suppress the pest population, which severely limits the application of mycopesticides in a cost-effective manner [117]. Environmental factors also limit the promotion of mycopesticides. For the BioMal^®^ mycoherbicide registered in Canada, is more effective in infection with dew 12–15 h after spraying, or rain greater than 6 mm within 48 h, and a temperature of about 20 °C [118].

### 4.2. Development Trend

#### 4.2.1. Virulence Enhancement by Genetic Engineering

Genetic engineering techniques can be employed to breed highly virulent strains. The first step involves identifying the target gene, which can be obtained from pathogenic fungi or biological toxin genes. Subsequently, a highly active promoter is screened to effectively express the target gene, followed by the construction of a high-efficiency transformation system. Genetic engineering is the most versatile method in strain manipulation, caused by adding either pathogenicity determining genes, stress resistance genes, or other factor that can increase the applicability of mycopesticides [117]. Currently, widely used methods include protoplast transformation, restriction enzyme-mediated transformation, electroporation, particle gun, and the *agrobacterium*-mediated method [119,120]. Fang et al. utilized the agrobacterium-mediated method to overexpress the chitinase Bbchit1 in engineered fungus of *B. bassiana*, thereby enhancing its virulence against aphids [121]. Wang and St Leger employed genetic engineering to transfer the insect-specific neurotoxin AaIT from scorpion *Androctonus australis* into *M. anisopliae*, resulting in the development of a highly virulent strain. This strain exhibited a 22-fold increase in toxicity towards tobacco hornworm *Manduca sexta* and a 9-fold increase in virulence towards adult yellow fever mosquitoes *Aedes aegypti* [122]. *B. bassiana* expressed the fire ant *Solenopsis invicta* pyrokinin β-neuropeptide (β-NP). the lethal dose (LD_50_) and lethal time (LT_50_) of the fungus to kill the target red fire ants were reduced, which was host specific [123]. However, there is some controversy with this approach. There is concern that enhanced fungal virulence through genetic engineering could affect non-target or beneficial insects.

#### 4.2.2. Fermentation Improvement

The fermentation process has a significant impact on the activity and virulence of fungi. Liquid fermentation is characterized by its fast speed, short cycle, and high yield. However, it is not resistant to storage, and blastospores have poor stress resistance. Higher quality blastospores can be obtained by improving culture conditions. Mascarin et al. found that at appropriate carbon nitrogen ratios, high glucose titers, and aeration rates, *B. bassiana* could achieve higher blastospore yields with low hyphal growth by liquid fermentation [124]. *B. bassiana* and *Cordyceps fumosorosea* produced higher concentrations of blastospores by using fermentation media containing more economical cottonseed flour than acid hydrolyzed casein as the nitrogen source. Moreover, blastospores of *B. bassiana* and *C. fumosorosea* killed whitefly nymphs faster and required lower concentrations compared with aerial conidia [125]. *C. fumosorosea* produced high concentrations and desiccation tolerant blastospores in a liquid medium containing 80 g/L glucose and 13.2 g/L casamino acids [126].

In solid state fermentation, there is no free water in the whole fermentation system, and the solid substrate provides carbon source, nitrogen source, water, and inorganic substances required for fungal growth. Solid substrates can be agricultural crops, agro-industrial residues, or inert materials impregnated with nutrients. Solid state fermentation takes advantage of low-cost agricultural residues, resolves the problem of solid waste disposal, and has higher fermentation productivity, as well as a higher end-concentration of products, higher product stability, and lower catabolic repression. Solid state fermentation produces aerial conidia, but it has a longer fermentation cycle [127,128]. Solid state fermentation is the most commonly used fermentation process for the production of mycopesticides due to its low cost and easy large-scale production of aerial conidia [129]. Zhang et al. found that the solid-state fermentation of *Trichoderma Brev* T069 was based on agricultural waste cassava peels, and the fermentation parameters were optimized by Response Surface Methodology (RSM). The spore production of *Trichoderma* was 9.31 × 10^9^ spores/g at 3rd days [130].

Another approach is solid–liquid two-phase fermentation, which combines the advantages of both liquid and solid-state fermentation. It involves first producing a large amount of highly active mycelium and blastospore through liquid fermentation and then transferring them to a solid medium to produce aerial conidia. This method reduces the fermentation period and enhances the stress resistance of conidia. However, the whole process increases the production cost of mycopesticides, and the risk of pollution also increases. At present, the main production process of *B. bassiana* products is solid–liquid two-phase fermentation [129].

#### 4.2.3. Combined Use

Mycopesticides have a prolonged period of infection, and when combined with other pesticides, they can enhance insecticidal efficiency. The mixture of *B. bassiana* and the resistance inducer potassium silicate had a higher control effect on *Frankliniella schultzei* [131]. Additionally, the combination of *Akanthomyces attenuatus* and Botanical Insecticide matrine demonstrated a significant synergistic effect on *Megalurothrips usitatus* [132]. Moreover, *B. bassiana* showed good compatibility with the acaricide pyridaben, and their interaction can be utilized for controlling *Tetranychus cinnabarinus* eggs effectively [133]. The combined treatment at the high dose of the entomopathogenic fungus *B. bassiana* and the entomopathogenic nematode Steinernema carpocapsae resulted in higher mortality rates in pests of the stored grains *Tribolium castaneum*, *Trogoderma granarium*, *Oryzaephilus surinamensis*, *Sitophilus oryzae*, *Rhyzopertha dominica*, and *Cryptolestes ferrugineus* compared with single treatments [134]. Rizwan et al. found that *B. bassiana* and diatomaceous earth (DE) are more effective in combination against *Tribolium castaneum* on wheat, which is due to the ability of DE to destroy the insect cuticle [135,136]. Wakil et al. demonstrated that simultaneous use of *B. bassiana* and the chemical insecticide spinetoram significantly reduced onion thrip larvae and adults and increased onion production [137].

## 5. Conclusions and Prospects

From the perspective of the number of mycopesticides applied worldwide, the proportion of mycopesticides is still very low. In recent years, more fungal strains have been registered in various countries. However, mycopesticides face certain limitations such as unstable control effects, low toxicity, slow effectiveness, strain degradation, and susceptibility to environmental influences. These limitations hinder their widespread application and development. At present, aerial conidia are the main active ingredients of mycopesticides. But the contamination of fungi during fermentation also limits their large-scale application. To overcome these challenges, it is essential to develop more mycopesticide formulations that can adapt to different environments. Additionally, genetic engineering can be utilized to breed highly virulent strains of fungi. Improving the culture conditions of spores can increase the spore yield and enhance the infection ability. other pesticides compatible with fungi can be screened for formulation. Furthermore, there is a need to expedite the translation of laboratory research findings into practical applications and create a market for mycopesticides.

## Figures and Tables

**Figure 1 jof-09-00940-f001:**
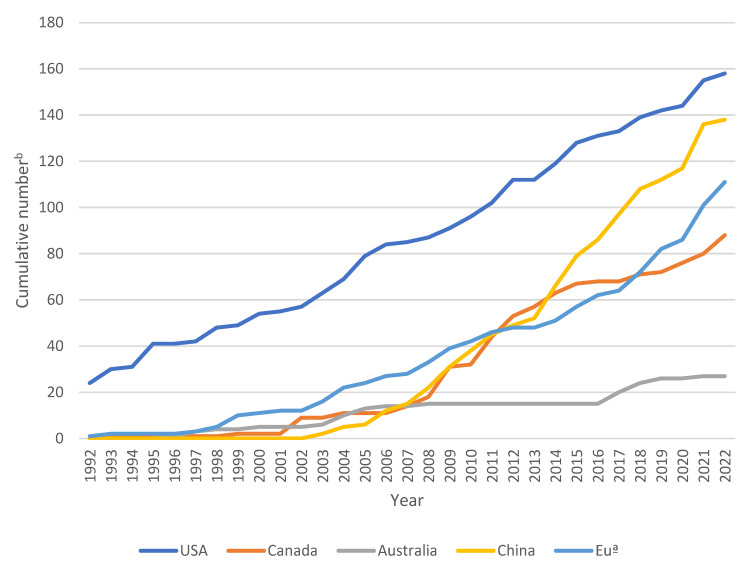
The cumulative evolution of mycopesticide products in the United States, Canada, Australia, China, and the European Union from 1992 to 2022. Mycopesticide products are on the rise in various countries. For the same active ingredients, the product names may be different in the EU member states, and the final statistical results are not very accurate. **^a^** Mycopesticide products in the 13 EU member states (The Netherlands, France, Portugal, Poland, Spain, Germany, Denmark, Greece, Italy, Austria, Belgium, Cyprus and Norway) were counted. Different member states of the EU choose the earliest registration year for the same product. ^b^ If the product is currently cancelled, the product is still counted at that time.

**Table 1 jof-09-00940-t001:** The registration and target of mycoinsecticides.

Mycoinsecticides ^a^	Country/Region ^b^ Where Approved/Registered	Target(s)
*Akanthomyces muscarius* Ve6 (formerly *Lecanicillium muscarium*)	EU, CA	Whiteflies, thrips
*Beauveria bassiana*	CHN, AUS	Rice leaf folder *Cnaphalocrocis medinalis*, aphids, termites
*Beauveria bassiana* strain 147	EU	*Paysandisia archon,* *Rhynchophorus ferrugineus*
*Beauveria bassiana* strain 203	EU	*Rhynchophorus ferrugineus*
*Beauveria bassiana* strain 447	USA	Ants
*Beauveria bassiana* strain ANT-03	USA, CA	Foliar-feeding pests and certain grubs
*Beauveria bassiana* strain ATCC 74040	USA, EU	Ants, aphids, armyworms, whiteflies
*Beauveria bassiana* strain CFL-A	CA	Annual bluegrass weevil larvae *Listronotus maculicollis*, asiatic garden beetle *Maladera castanea*
*Beauveria bassiana* strain GHA	USA, EU, CA	Scarab beetles, leaf-feeding beetles, whiteflies, aphids, thrips
*Beauveria bassiana* strain HF23	USA, CA	Houseflies
*Beauveria bassiana* strain PPRI 5339	USA, EU, CA	Certain piercing, sucking, and chewing pests (insects and mites)
*Beauveria bassiana* strain R444	CA	Black cutworm, corn flea beetle, nematodes
*Beauveria bassiana* strain IMI389521	EU	Coleoptera pests *Oryzaephilus surinamensis, Sitophilus granaries, Cryptolestes ferrugineus*
*Beauveria bassiana* strain NPP111B005	EU	*Cosmopolites sordidus, Rhynchophorus ferrugineus*
*Beauveria bassiana* strain ZJU435	CHN	Fall armyworm *Spodoptera frugiperda,* whitefly *Trialeurodes vaporariorum*
*Conidiobolus major*	CHN	Whiteflies, aphids
*Cordyceps javanica* Ij01 (formerly *Isaria javanica*, *Paecilomyces javanicus*)	CHN	*Spodoptera litura* Fabricius
*Cordyceps javanica* JS001 (formerly *Isaria javanica*, *Paecilomyces javanicus*)	CHN	Whitefly *Bemisia tabaci*
*Cordyceps fumosorosea* strain Apopka 97 (formerly *Isaria fumosorosea*, *Paecilomyces fumosoroseus*)	USA, EU	Whiteflies, thrips, aphids, spider mites
*Cordyceps fumosorosea* strain FE 9901 (formerly *Isaria fumosorosea*, *Paecilomyces fumosoroseus*)	USA, EU, CA	Aphids, weevils, whiteflies
*Metarhizium anisopliae*	CHN	Thrips, locusts, *Carposina niponensisi*, *Spodoptera exigua*
*Metarhizium anisopliae* strain CQMa421	CHN	*Chilo suppressalis, Spodoptera frugiperda*
*Metarhizium anisopliae* strain ESF1	USA	Termites
*Metarhizium acridum* (formerly *Metarhizium anisopliae* var. acridum)	AUS	Australian plague locust—nymphs, grasshoppers
*Metarhizium brunneum* strain Ma 43 (formerly *Metarhizium anisopliae* var. anisopliae)	EU	Japanese beetle *Popillia japonica*, Garden chafer *Phyllopertha horticola*, Summer chafer *Amphimallon solstitialis*, European chafer *Amphimallon majalis*
*Metarhizium brunneum* strain F52 (formerly known as *Metarhizium anisopliae* strain F52)	USA, CA	Mites, thrips, ticks, weevils and whiteflies
*Nosema locustae*	USA, CA, CHN	Grasshoppers, Mormon cricket

^a^ Current names according to the database Index Fungorum http://www.indexfungorum.org/. ^b^ United States of America (USA), Australia (AUS), Canada (CA), China (CHN), European Union (EU). Source: (USA) https://ordspub.epa.gov/ords/pesticides/f?p=chemicalsearch:1, (AUS) https://portal.apvma.gov.au/pubcris, (CA) http://pr-rp.hc-sc.gc.ca/ls-re/result-eng.php?p_search_label, (CHN) http://www.chinapesticide.org.cn/, (EU) https://food.ec.europa.eu/plants/pesticides_en (accessed on 22 July 2023).

**Table 2 jof-09-00940-t002:** The registration and target of mycofungicides or nematophagous fungi.

Mycofungicides or Nematophagous Fungi	Country/Region Where Approved/Registered	Target(s)
*Ampelomyces quisqualis* strain AQ10	USA, EU	Powdery mildew
*Aspergillus flavus* strain AF36	USA	Strains of the fungus *Aspergillus flavus* that produce aflatoxin
*Aspergillus flavus* strain NRRL 21882	USA	Strains of the fungus *A. flavus* that produce aflatoxin
*Aureobasidium pullulans* strains DSM 14940 and DSM 14941	USA, EU, CA, AUS	Bacterial and fungal flower and foliar diseases
*Candida oleophila* isolate I-182	USA	Post-harvest fungicide
*Candida oleophila* strain O	USA, EU	For post-harvest control of gray mold *Botrytis cinerea* and blue mold *Penicillium expansum*
*Clonostachys rosea* strain CR-7	USA	*Botrytis*, *Colletotrichum*, *Monilinia*, *Sclerotinia*, *Alternaria*, *Fusarium*, and *Didymella*
*Clonostachys rosea* strain J1446	USA, EU, CA	Seed borne and soil borne fungi, such as *Fusarium*, *Pythium* and *Phytophtora,* foliar fungal diseases
*Paraphaeosphaeria minitans* (formerly *Coniothyrium minitans*) strain CON/M/91-08	USA, EU, CA	*Sclerotinia* spp.
*Paraphaeosphaeria minitans* (formerly *Coniothyrium minitans*) strain ZB-1SB	CHN	*Sclerotinia* spp.
*Paraphaeosphaeria minitans* (formerly *Coniothyrium minitans*) Campbell CGMCC8325	CHN	*Sclerotinia* spp.
*Duddingtonia flagrans* strain IAH 1297	USA	Nematodes
*Gliocladium virens* GL-21	USA	Fungi that cause “damping off” disease and root rot.
*Muscodor albus* strain QST 20799	USA	Bacteria, fungi, and nematodes
*Muscodor albus* strain SA-13	USA	Soil-borne plant diseases and plant-parasitic nematodes
*Metschnikowia fructicola* strain NRRL Y-27328	USA, EU	*Monilinia fructigena, Monilia laxa, Botrytis cinerea*
*Myrothecium verrucaria* dried fermentation solids and solubles	USA	Nematodes
*Purpureocillium lilacinum* [formerly *Paecilomyces lilacinus* (Thom) Samson]	CHN	Root-knot nematodes *Meloidogyne* spp.
*Purpureocillium lilacinum* strain 251 (formerly *Paecilomyces lilacinus* strain 251)	USA, EU	Root-knot nematodes *Meloidogyne* spp., cyst nematodes *Geterodera* spp. and *Globodera* spp.
*Purpureocillium lilacinum* strain PL 11	USA, EU	Root-knot nematodes *Meloidogyne* spp.
*Pseudozyma flocculosa* strain PF-A22 UL	USA	Powdery mildew
*Pseudozyma flocculosa*	CA	Soil-borne diseases caused by fungus
*Phlebiopsis gigantea* strain VRA 1992	USA, CA	*Heterobasidion* spp.
*Phlebiopsis gigantea* strain VRA 1835, VRA 1984 and FOC PG 410.3	EU	*Heterobasidion* spp.
*Saccharomyces cerevisiae* extract hydrolysate	USA	Bacterial diseases
*Saccharomyces cerevisiae* strain LAS02	EU	Storage diseases *Monilinia* spp., *Botrytis cinerea*
*Trichoderma asperellum* strain ICC 012	USA, EU, CA	Fungal soil diseases in vegetables and ornamentals
*Trichoderma asperellum* strain T25	EU	*Phythophthora* sp. *Fusarium* sp. *Pythium* sp.
*Trichoderma asperellum* strain TV1	EU	*Pythium* spp. *Rhizoctonia* spp. *Fusarium* spp.
*Trichoderma asperellum* strain T34	USA, EU, CA	*Fusarium oxysporum* f.sp. *dianthi*
*Trichoderma asperelloides* strain JM41R	USA	*Rhizoctonia* spp. *Fusarium* spp.
*Trichoderma atroviride**strain* SC1	USA, EU	Wood and canker diseases
*Trichoderma atrobrunneum* (formerly *Trichoderma harzianum)* strain ITEM 908	EU	*Pythium* spp., *Rhizoctonia* spp., *Fusarium* spp.
*Trichoderma atroviride* strain IMI 206040	EU	*Pythium* spp., *Rhizoctonia* spp., *Fusarium* spp.
*Trichoderma atroviride* strain T11	EU	*Pythium* spp., *Rhizoctonia* spp., *Fusarium* spp.
*Trichoderma atroviride* strain I-1237	EU	Wood decay diseases
*Trichoderma gamsii* strain ICC 080	USA, EU, CA	Fungal soil diseases in vegetables and ornamentals
*Trichoderma harzianum*	CHN, AUS	Clubroot disease, *Botrytis cinerea, Rhizoctonia* spp., downy mildew
*Trichoderma harzianum* LTR-2	CHN	Brown spot, grey mould *Botrytis cinerea*
*Trichoderma harzianum* DS-10	CHN	Grey mould *Botrytis cinerea*
*Trichoderma harzianum* T-39	USA	*Botrytis cinerea*
*Trichoderma harzianum* strain T78	USA	*Fusarium*, *Phytophthora* spp., *Pythium* spp., *Rhizoctonia*, *Sclerotium* spp.
*Trichoderma hamatum* isolate 382	USA	Diseases caused by soil borne plant pathogens
*Trichoderma harzianum* rifai strain T-22	USA, EU, CA	Various fungi that cause seed rot, diseases of plant roots, and other plant diseases
*Trichoderma harzianum* rifai strain KRL-AG2	USA, CA	Root pathogens in greenhouse tomatoes, cucumbers, and ornamentals
*Trichoderma polysporum* ATCC 20475	USA	Fungi that infect tree wounds
*Trichoderma viride* ATCC 20476	USA	Fungi that infect tree wounds
*Trichoderma virens* strain G-41	USA, CA	Fungal soil diseases in vegetables, ornamentals
*Trichoderma* spp.	CHN	Various fungi that cause seed rot, diseases of plant roots, and other plant diseases
*Typhula phacorrhiza* strain 94671	USA, CA	Snow molds in turf
*Ulocladium oudemansii* strain U3	USA	*Botrytis cinerea* and *Sclerotinia sclerotiorum*
*Verticillium dahliae* strain WCS850	USA, EU, CA	Dutch elm disease
*Verticillium chlamydosporium* Goddard	CHN	Root-knot nematodes

**Table 3 jof-09-00940-t003:** The registration and target of mycoherbicides.

Mycoherbicides	Country/Region Where Approved/Registered	Target(s)
*Alternaria destruens* strain 059	USA	Dodder *Cuscuta* spp.
*Chondrostereum purpureum* strain PFC 2139	USA, CA	Inhibits the sprouting and regrowth of shrubs and hardwood trees
*Chondrostereum purpureum* strain HQ1	USA	Inhibits the sprouting and regrowth of shrubs and hardwood trees
*Colletotrichum gloeosporioides* f. sp *aeschynomene*	USA	Northern jointvetch *Aeschynomene virginica*
*Phoma macrostoma*	CA	Broadleaved weeds like dandelion, Canada thistle, and clover
*Phytophthora palmivora* MWV	USA	*Morenia orderata*, commonly known as strangler vine or milkweed vine
*Puccinia thlaspeos* strain woad (dyer’s woad rust)	USA	Dyer’s woad
*Lasiodiplodia pseudotheobromae* NT039, *Macrophomina phaseolina* NT094, *Neoscytalidium novaehollandiae* QLD 003	AUS	*Parkinsonia* spp.

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
