# Peer review of "The Registration Situation and Use of Mycopesticides in the World"

_jof, 2023, doi:10.3390/jof9090940_

Round 1
Reviewer 1 Report
This review article is interesting and well written. Since a historical perspective on the uses of mycopesticides is also included, I suggest to add this to the title: The registration situation and use of mycopesticides in the world.
Below some specific comments:
Line 66: and there are also nematophagous fungi such as Purpureocillium lilacinum
Line 82: Earlier than? Text is not clear here since the US was first registered first in 1991 and the former Soviet Union in 1965. Same thing in lines 137-140. It seems to be same information repeated twice, please clarify or unify.
Line 157: Is Beauveria brongniartii not registered? That is understood by what line 151 says. Thus, B. brongniartii should be not include.
3.1.3: Is Paecilomyces registered? (does not appear in Table 1), or just clarify that it is included as nematophagous fungus. Also, if Paecilomyces is now Isaria, then it should be listed in section 3.1.4.
3.1.5: Please first define that Akanthomyces lecanii was first named as Lecanicillium in line 197.
Line 210: Which bacteria?
Table 2: First column header should be nematophagous
Line 326: it took….….(lower case) Also, remove +0.5 ℃.
Line 342: Be careful here, slow and low virulence compared to? Maybe it would be better to delete this sentence and start the parapraph with “Genetic engineering…..”
Line 349: Fang et al.
4.2.1: This section does not make much sense in the context of registration, unless the authors clarify the additional drawbacks of registering GMOs.
360: in solid state….(lower case)
363: Solid state (upper case)
4.2.2: Is it known how many products registered worldwide include fungi produced by solid, liquid or two-phase fermentations? It would be interesting to include this information, otherwise this section seems out of context.
English Language is correct.
Reviewer 2 Report
In the Review entitled "The registration situation of mycopesticides in the world"
the authors seek to provide a general overview of the history of the development and current status of application and registration of mycopesticides worldwide. Mycopesticides are living preparations that utilize fungal cells, such as spores and hyphae, with roles as insecticides, fungicides, herbicides, and nematophagous fungi. Mycopesticides account for about 10 percent of the biopesticide market.
The authors address issues due to different registration in different states-in fact, it took a long time before fungi were officially recognized and registered as pesticides. About 1400 biopesticide products and 1000 active ingredients have been registered worldwide.
The paper is clear and well organized, it turns out to be somewhat didactic but useful to have an overview of the current situation of the use of mycopesticides, whose role is certainly of extreme importance in the field of biological control and therefore safety for humans and the environment.
The work has small typos :
Table 1, 2 , 3 and in the text: correct the word "registration"
in the text: correct the word "nematophungus"
line 360: change "In solid "to "in solid"
Minor editing of English language required
Reviewer 3 Report
This review is an interesting article and the authors intended to bring the updated information on different types of mycopesticides, their history and data on the evolution of mycopesticides in the modern world. The authors albeit touches briefly the combination potential of mycopesticides but there is a room to strengthen this important aspect in the review article. The problems and advanced development trends has also been briefly given. This review is timely and will attract good readership. Based on the importance of the article I am of the opinion that this review article may be accepted after minor revision for its publication in Journal of Fungi.
Some of my comments are as follows to be considered by authors for revision:
- I may indicate some grammatically weak sentences and minor spelling mistakes, in my opinion the authors should carefully look and improve the language/spellings for easy understanding of readers
- The Objectives both in Abstract and Introduction section should be consistent and there should be no contradiction
- Its better to provide schematic presentation (original) indicating in detail the mode of infection of mycopesticides separately for each type
- Sub-section 4.1.2 and 4.1.3 should be further strengthened
- Sub-section 4.2.3. “Combined use” need attention of authors as the recent literature on the combination of mycoinsecticides with other environment friendly agents (microbes etc.) and even insecticides is published in recent years. I am indicating some examples and I suggest authors should consider them to include in this review article (https://www.mdpi.com/2073-4395/12/5/1160; https://doi.org/10.3390/jof9080835; https://doi.org/10.3390/pathogens12060773; https://doi.org/10.1002/ps.7503; https://doi.org/10.3390/agronomy12081928; https://doi.org/10.3390/agronomy12051160; https://doi.org/10.1002/ps.6899; https://doi.org/10.1093/jee/toaa209; https://doi.org/10.1111/1748-5967.12260 and many others you may browse)
- Conclusion and Prospects: the authors may further extend the possibilities and obstacles with reference to scalability of entomopathogenic fungi? One important issue is contamination of these products by other microbes that can impact end-user confidence, what is known about this issue?
- The economics of the mycopesticides/mycoinsecticides should be briefly discussed in this review article
I may indicate some grammatically weak sentences and minor spelling mistakes, in my opinion the authors should carefully look and improve the language/spellings for easy understanding of readers
